# Natural Polymeric Compound Based on High Thermal Stability Catechin from Green Tea

**DOI:** 10.3390/biom10081191

**Published:** 2020-08-16

**Authors:** Malgorzata Latos-Brozio, Anna Masek

**Affiliations:** Institute of Polymer and Dye Technology, Lodz University of Technology, Stefanowskiego 12/16, 90-924 Lodz, Poland; malgorzata.latos-brozio@dokt.p.lodz.pl

**Keywords:** natural polyphenols, (+)-catechin, polymerization, antioxidant properties, thermal analysis

## Abstract

Catechin is a plant polyphenol with valuable antioxidant and health-promoting properties. Polymerization is one way to stabilize flavonoids and may cause changes in their specific properties. The aim of this study is to obtain a polymeric complex catechin compound with high thermal stability. As a result of polymerization, a condensed and cross-linked catechin structure was obtained, which guaranteed high thermal resistance and, moreover, the phosphorus groups added in the second step of polymerization ensured that the compound obtained had thermal stability higher than natural condensed tannins. The first step of self-polymerization of (+)-catechin may be an easy way to obtain proanthocyanidins with greater antioxidant activity. The second step of the polymerization obtained a polymeric complex catechin compound that showed better thermal stability than catechin. This compound can potentially be used as a new pro-ecological thermal stabilizer.

## 1. Introduction

Polyphenols are widely distributed in the plant kingdom and they are one of the most important groups of secondary metabolites of plants [1,2]. An important and well-studied group of polyphenols are flavonoids, which consist of two phenolic rings and an oxygenated heterocycle. Flavonoids are a very large group of about 6000 compounds, which are divided into many classes, such as anthocyanins, flavonols, flavanols, flavones, and chalcones [3,4].

Catechins are included in the subgroup flavan-3-ols or flavanols and they are categorized into two groups—free catechins and esterified catechins. Galocatechin, epicatechin, and epigallocatechins belong to the group of non-esterified catechins. Epigallocatechin gallate, epicatechin gallate, gallocatechin gallate, and catechin gallate are classified as esterified catechins [5,6,7]. Catechins are present in high amounts in leaves of green tea. One gram of dried green tea leaves contains about 200 mg catechins. The total content of catechins varies significantly depending on the species, variety, cultivation location, season, illumination and altitude. The main catechins found in green tea are (−)-epigallocatechin gallate, (−)-epicatechin, (−)-epigallocatechin, (−)-epicatechin gallate, and (+)-catechin [8].

Catechins, like other flavonoids, exhibit various biological properties for the prevention of diseases, e.g., chemopreventive activity, antibacterial activity, antiproliferative activity, and antioxidative effect [9,10]. In nature, catechins occur in many oligomeric structures, primarily condensed tannins (also known as proanthocyanidins) [11]. In clinical practice, oligomeric flavonoids are more popular because the monomeric forms of flavonoids are limited by instability when exposed to light, heat, and basic conditions; poor bioavailability; rapid metabolism; and poor membrane permeability [12,13,14,15,16,17]. Furthermore, condensed tannins derived from natural origin often require costly treatments to achieve the desired purity, and exhibit significant variability, leading to inconsistent behavior in potential applications [18,19,20,21].

Flavonoids can be modified and stabilized by polymerization [22]. The beneficial properties of flavonoids, such as bactericidal or antioxidant activity, may be intensified as a result of the polymerization of these compounds [23]. In the literature, enzymatic polymerization is the most widely presented method of polymerization of natural polyphenols. The enzymatic polymerization reactions catalyzed by laccase [24] and also by horseradish peroxidase (HRP) [25] allowed to obtain oligomeric catechin structures. Polycondensation of catechins with aldehydes in the presence of an acid catalyst is another method of obtaining poly(catechin) [26,27,28]. Stronger properties such as superoxide scavenging capacity, inhibition of the xanthine oxidase (XO) enzyme and peroxidation of low-density lipoprotein (LDL) have been correlated with the higher molecular weight of poly(catechin) [27]. However, studies showed that the reducing power of the poly(catechin) was weaker than the monomeric structure, with inverse dependence on molecular weight. The method of obtaining poly(catechin) is also the preparation of oligomers and polymers from catechin and quercetin, using only a solvent and hydrochloric acid. The procedure allows for obtaining water-soluble catechin polymers up to 3300 g/mol and water-soluble quercetin polymers up to 3100 g/mol, as well as polymeric catechin structures with higher molecular weight up to 1.5 × 10^4^ g/mol. The authors showed that oligomers of catechin were characterized by good antimicrobial efficacy against Gram-positive and Gram-negative bacteria. Evidence of the destruction of bacterial cells by poly(catechin) was demonstrated on the basis of SEM images and tomography [29]. In addition, scientific literature also indicates the possibility of obtaining catechin complexes with metal ions, including with Al^3+^; Cu^2+^; Ni^2+^; Mg^2+^; Mn^2+^; Zn^2+^, and Fe^3+^ [30]. However, to date, no method has been described for the preparation of catechin complex polymeric compounds containing phosphate salt derivatives having high thermal stability, and this is the scientific novelty of this manuscript.

The aim of the study is to obtain a polymeric complex catechin compound with high thermal stability as a result of self-polymerization and photopolymerization. According to literature data, poly(catechin) can be obtained by photopolymerization of catechin in an alkaline environment by irradiation with light from the blue range. The resulting product was identified as proanthocyanidins [31]. However, Liang and co-authors [31] found no self-polymerization of catechin in the dark at pH 6, 7, and 8. They only proved that catechin polymerizes when exposed to blue light in an alkaline environment (pH 7 and 8). Catechin dimer (poly(catechin)) has the molecular formula C_30_H_26_O_12_ and exact mass of 578.14 g/mol. The total phenolic content and the antioxidant activity (determined by O_2_^−^ scavenging activity) of proanthocyanidins compared to those of catechin did not significantly change [31].

The proposed self-polymerization of (+)-catechin in phosphate buffer at pH 7 or 8 in this manuscript may be a new and easy way to obtain proanthocyanidins. In addition, the reaction can be accelerated by exposure to light from the blue range. Moreover, the second step of the proposed polymerization allows a polymeric complex catechin compound with significant thermal stability to be obtained. Based on the literature review, it was found that this catechin-based compound has not yet been described and is a scientific novelty. The compound can potentially be used as a pro-ecological stabilizer, e.g., for environmentally friendly materials with wide application.

## 2. Materials and Methods

### 2.1. Polymerization of (+)-Catechin

Step 1: 1 mM catechin ((+)-catechin hydrat ≥98% HPLC, MW: 290,27 g/mol, Sigma-Aldrich, Hong Kong, China) was dissolved in phosphate buffer Na_2_HPO_4_-KH_2_PO_4_ (0.1 M, CHEMPUR, Piekary Slaskie, Poland) at pH 6, 7, 8. The solutions were irradiated with blue light (440–490 nm) using a Bio Light BL100 lamp (BIOMAK, Piastow Poland), and the irradiance was controlled using a TM-206 solar irradiance meter (Tenmars Electronics Co., Taipei, Taiwan). The radiation intensity during the reaction was 2.0 W/m^2^. The solutions were irradiated from 2 h to 24 h. Solutions left in the dark (from 2 h to 216 h, room temperature), not exposed to blue light, were used as reference tests.

Step 2: During the polymerization, significant precipitation was observed, therefore, after the polymerization reaction, the precipitates were decanted and the sediment in the remains of pH 7 or 8 phosphate buffer (1/10 of the initial buffer volume) was dried at 55 °C for 48 h (dryer with forced convection Solid Line FD-S 56, Binder, Tuttlingen, Germany).

### 2.2. Analysis of the Kinetics of the Polymerization Reaction

Polymerization reactions were monitored using a UV-Vis (Ultraviolet–visible) spectrophotometer (Evolution 220, Thermo Fisher Scientific, Waltham, MA, USA), recording spectra at wavelengths of 190–1100 nm. Phosphate buffers with appropriate pH were used as the reagent blanks.

### 2.3. Infrared (FTIR) Spectroscopy

A Nicolet 670 FTIR (Fourier-transform infrared spectroscopy) spectrophotometer (Thermo Fisher Scientific, Waltham, MA, USA) was used to analyze the poly(catechin) structure. Samples of reference (+)-catechin and poly(catechin), in the form of dried powders were placed at the output of infrared beams. As the result of the test, oscillating spectra were obtained, the analysis of which allows determination of the functional groups with which the radiation interacted.

### 2.4. Hydrogen Peroxide Scavenging Capacity

The ability of the (+)-catechin and poly(catechin) to scavenge hydrogen peroxide was determined according to the method developed by Ruch et al. [32,33]. A solution of hydrogen peroxide (40 mM) was prepared in phosphate buffer at pH 7.4 (1 M, Sigma-Aldrich, St. Louis, MO, USA). Catechin reference solutions (1 mM) in phosphate buffers (0.1 M) at pH 6, 7, and 8 were prepared. The standard catechin buffer solutions were added to the hydrogen peroxide solution (0.8 mL, 40 mM) 30 min after their preparation, after 216 h in the dark, and after 24 h irradiation with blue light. Absorbance of hydrogen peroxide at 230 nm was determined 15 min later against a blank solution containing the phosphate buffer without hydrogen peroxide. The percentage of hydrogen peroxide scavenging of catechin and poly(catechin) was calculated as follows (1):%Scavenged[H_2_O_2_] = [(AC–AS)/AC] × 100(1)
where AC is the absorbance of the control and AS is the absorbance in the presence of the sample of catechin or poly(catechin).

Three measurements were made for each sample. The results are shown as mean ± standard deviation (SD).

### 2.5. Thermal Stability of (+)-Catechin and Pol (Catechin)

The thermogravimetric (TG) analysis of (+)-catechin and poly(catechin) was performed using a Mettler Toledo Thermobalance (TA Instruments, Greifensee, Switzerland). Samples of approximately 10 mg were placed in aluminum pans and heated from 25 °C to 800 °C under argon flow (50 mL/min). A heating rate of (5 °C/min) was used.

### 2.6. Differential Scanning Calorimetry (DSC) of (+)-Catechin and Poly (Catechin)

The temperature ranges of the samples’ phase changes were determined using a Mettler Toledo DSC analyser (TA 2920; TA Instruments, Greifensee, Switzerland). The samples (5–6 mg, placed in 100 μL aluminium crucibles) were heated from −80 to 400 °C at a rate of 10 °C/min in air.

### 2.7. Digital Microscopy

Photos of powders were made using a digital microscope VHX-7000 at magnifications of 20–300 times.

## 3. Results and Discussion

Natural phenolic compounds are stable in an acidic environment but not stable in neutral and alkaline aqueous solutions for esterification or molecular rearrangement. Catechins are sensitive to heat and light. The catechin heterocyclic ring was opened by UV-C radiation for 20 h [34]. Literature data [31] have shown that catechin is unstable with blue light illumination in neutral and alkaline aqueous solvent but is stable with green and red light illumination.

Figure 1 shows the effects of the first step of catechin polymerization. Figure 1A summarizes the UV-Vis spectra of catechin solutions in phosphate buffers at pH 6, 7, and 8; left in the dark from 2 h to 216 h; and irradiated with blue light (440–490 nm) from 2 h to 24 h. The research was carried out until a clear precipitation occurred, which prevented proper UV-Vis measurements. Based on the analysis of UV-Vis spectra of catechin solution in buffer at pH 6 (Figure 1A1,A2), the appearance of an additional peak with a maximum at 400 nm was found. The clear peak appeared until after 144 h of reaction in the dark and 18 h of exposure to blue light. Due to pronounced organoleptic changes (turbidity, color change to slightly yellow) of the solution irradiated for 8 h with blue light, it was decided to continue the reaction, although no visible changes were observed on UV-Vis spectra. Figure 1A3,A4 also summarize the UV-Vis spectra of catechin solutions prepared in buffer at pH 7. The appearance of the additional peak in the range 370–470 nm was observed in the UV-Vis spectra. A clear peak was identified after 24 h reaction in the dark and after 8 h blue light exposure. Similar results were also obtained for catechin solutions in buffer at pH 8 (Figure 1A5,A6). Based on UV-Vis spectra, an additional peak in the range of 370–470 nm was found. A clear peak appeared after 24 h reaction in the dark and after 4 h of blue light exposure. The jagged spectra in Figure 1A5,A6 illustrate precipitation of buffer solutions at pH 8.

Based on the available literature data, the peak in the range of 370–470 nm is a characteristic peak for proanthocyanidin or oligomeric forms of catechin. Nagarajan et al. [35] interpreted the peak in the range of 300–500 nm as a characteristic peak for oligomeric forms of catechin, while Liang et al. [31] reported that the peak in the range of 350–550 nm corresponds to proanthocyanidin. Proanthocyanidin was also confirmed by HPLC analysis.

The peak present in the range 370–470 nm in the spectra obtained confirmed the possibility of photopolymerization of catechin in pH 6-8 environment, and especially at pH 7 and pH 8. Along with the increase in the basic nature of the reaction environment, the speed and intensity of the appearance of the polymerization product increased.

Liang and co-authors [31] described the possibility of catechin polymerization only under the influence of blue light irradiation in an alkaline environment (pH 8). The authors ruled out polymerization of this polyphenol in the dark in phosphate buffer at pH 8. Unlike available literature data, in studies presented in this manuscript, it was surprising that the catechin could be polymerized in an environment with only adequate pH and no radiation. Polymerization of catechin, in the dark, can be called a self-polymerization reaction in a buffer solution. Moreover, it has been shown that blue light exposure accelerated and increased the intensity of catechin self-polymerization.

The kinetics of catechin photopolymerization and self-polymerization reactions were determined based on UV-Vis spectra (Figure 1B,C). Kinetics was defined as the ratio of the peak intensity characteristic of polymeric forms of catechin to time. Absorbance at 400 nm was assumed as the peak intensity for comparison. The correlation between the absorbance (400 nm) of the poly(catechin) peaks obtained in the dark (B1) corresponding to the poly(catechin) peaks obtained by irradiation with blue light (B2) is summarized in Figure 1C. Irradiation of buffer solutions with blue light significantly accelerated and increased the intensity of the catechin polymerization reaction. Irradiation of catechin solutions with blue light caused a six-fold increase in the rate of polymerization reaction in buffers at pH 6 and 7, and a nine-fold increase in the polymerization of catechin in a phosphate buffer environment at pH 8. As indicated by the polymerization kinetics, the catechin’s polymerization capacity increased as the buffer pH increased (pH 6 < pH 7 < pH 8). Irradiation of solutions with blue light further intensified the polymerization reactions. Very good effects were achieved for self-polymerization of catechin in a buffer at pH 8 and even better for the same solution illuminated with light in the range 440–490 nm.

Many natural compounds are sensitive to visible light irradiation in the range from 400 nm to 800 nm. Catechin is unstable with blue light illumination in neutral and alkaline aqueous. Unlike green and red light, only blue light causes molecular degradation of (+)-catechin in neutral or alkaline aqueous solutions [31] and activates this compound for further reactions. Under these conditions, the catechin rings open and participate in the polymerization reaction. Therefore, the best and fastest polymerization effect was observed in buffers at pH 7 and 8 irradiated with blue light.

The formation of catechin dimeric forms was observed both during the irradiation of the solutions with blue light and under the reaction conditions in the dark. Catechin dimerization occurred via the loss of two hydrogen atoms, i.e., an oxidation process. The alkaline reaction environment may be the factor causing the oxidation of catechin in both reactions. Li et al. [5] showed that the stability and degradation of catechins from green tea concentrates was dependent, among others, from the pH of the environment. Green tea catechins are stable under acidic pH conditions. This behavior of catechins can be partly explained by the direct increase in the rate of oxidation with increasing pH. According to the literature data [5], pKa1 epigallocatechin gallate (EGCG) is 7.55 ± 0.03. Therefore, the ionization state of EGCG, which indicates the ability of the catechin to donate protons, may be a factor in causing catechin ring opening (leading to compound degradation). In a neutral and basic environment, epigallocatechin gallate (EGCG) showed an increased ability to oxidation and the formation of semiquinone free radical of EGCG [5].

Thus, in the reactions described in this manuscript, the oxidizing agent of (+)-catechin during the reaction in the dark and during photopolymerization may therefore be an alkaline pH, which results in an increased capacity of the catechin to oxidize and form free radicals. These factors contribute to the formation of the dimeric forms of catechin. During polymerization with blue light, the energy of light is additionally a factor accelerating and intensifying the process.

Due to the reaction environment (phosphate buffer), antioxidant activity was determined as hydrogen peroxide scavenging capacity (Figure 2C). Catechin solutions 30 min after preparation in buffers with a pH of 6 to 8 showed similar antioxidant activity, which was respectively: pH 6–28.3%, pH 7–32.3%, pH 8–33.3%. Buffer solutions at pH 6 and 7 containing polymerized forms of catechin exhibited slightly higher or higher antioxidant activity than reference samples. Only solutions with pH 8 with polymerized catechin were characterized by a very marked increase in the ability to scavenge H_2_O_2_ (216 h Dark–58.8%, 24 h Blue 62.9%). Buffer solutions at pH 6 and 7 containing polymerized forms of catechin may have a lower hydrogen peroxide scavenging capacity due to the lower content of poly(catechin). Based on UV-Vis spectra, it can be concluded that buffer solutions at pH 8 contain the largest amount of poly(catechin) and, thus, the largest number of active hydroxyl groups responsible for the highest antioxidant activity. The results showed that the polymeric forms of catechin had greater antioxidant activity than the monomeric catechin.

Liquid NMR analysis of phosphate buffer solutions after first step polymerization showed only the presence of buffer. Due to the low concentration of poly(catechin) in the buffer sample tested, it was not visible. Due to these results, a second polymerization stage was carried out. Poly(catechin) was decanted and dried. The obtained powders (only pH 7 and 8, pH 6 was too small) were prepared for liquid NMR studies; however, they were not soluble and did not swell in water, ethanol, toluene, acetone, chloroform, DMF and DMSO and their mixtures, which prevented liquid NMR analyses. Lack of solubility may indicate cross-linking or condensation of the polymeric polyphenol obtained, and significantly impedes structure analysis. The first stage of polymerization allowed proanthocyanidin to be obtained. In the second stage, reaction conditions (alkaline or neutral medium and elevated temperature) favoured the further reaction of forming condensed catechin structures and crosslinking and, moreover, forming bonds with phosphate groups. The polymeric form of the compound obtained, and the presence of group P-O-C were indicated by FTIR analysis and are discussed later in the manuscript.

In Figure 2A, the scheme of polymerization reaction is proposed. The first step consisted of obtaining proanthocyanidine as a result of self-polymerization of catechin in phosphate buffers and during irradiation of catechin buffer solutions with blue light. Irradiation of solutions with blue light intensified the production of proanthocyanidin. The second stage of polymerization consisted of decanting the buffers and leaving the sediments in 1/10 of the initial volume of buffer to dry at 55 °C. As a result of this polymerization step, polymeric complex catechin compound was obtained. Visual changes of (+)-catechin solution after preparation (B1); left 216 h in dark and irradiated with blue light for 24 h (B2); and after decantation of sediment and drying (B3) are shown in Figure 2B. Figure 2D summarizes the photos of poly(catechin) powders taken using a digital microscope with a magnification of 20 to 300 times. (+)-Catechin (D3) had a needle-like structure with needle length of about 20 μm to 50 μm. Catechin powders after polymerization had a different structure than the reference catechin. The structure of polymeric complex catechin compound obtained as a result of polymerization in the dark (D1) and under the influence of exposure to blue light (D2) in pH 8 showed a plate-like structure. The structure of polymeric complex catechin compound powders was homogeneous.

Insolubility of the powders obtained significantly hindered the determination of the structure of the compound. The structure of polymeric complex catechin compound powders was examined by FTIR spectroscopy and UV-Vis. Figure 3 shows FTIR spectra of (+)-catechin and poly(catechin) powders obtained as a result of polymerization in the dark in phosphate buffer and photopolymerization with blue light in solutions of pH 7 (A) and pH 8 (B), and UV-Vis spectra of catechin and poly(catechin) powders (C).

Polymeric catechin powders were combined with the monomeric form of this polyphenol. In the case of reactions carried out in buffer at both pH 7 and pH 8, the spectra of polymeric catechin powders obtained during polymerization in buffer only and with photopolymerization were identical. This suggests that the final products of catechin self-polymerization in an alkaline environment and photopolymerization with blue light are the same polymeric forms of catechin. According to literature data [36,37] absorption bands at 3390, 1602, and 1065 cm^−1^ were attributed to the characteristic functional groups of polyflavonoids. The bands in the range 1310–1390 cm^−1^ corresponded to phenol group vibrations -C-OH deformation vibration, and in the range 1112–1172 cm^−1^ to -C-OH stretching vibration [38]. The disappearance of hydroxyl groups at 3390 cm^−1^ and the appearance of bands in the ranges 2200–2600 cm^−1^ and 2720–2990 cm^−1^ may suggest cross-linking or condensation reactions of catechin, which probably occurred after centrifugation of the polymeric form of catechin from buffer solutions and drying of the sediments. Cross-linking or condensation of catechin may have caused the product of polymerization to be insoluble. These bands may correspond to the group Ar-OH, CH_3_, -O-CH_3_, and -O-CH_2_-O-. In the range of 810–950 cm^−1^, characteristic bands for the C-O-C bands appeared. In addition, in the range 1090–1000 cm^−1^ there were characteristic bands for P-O-C [39].

Figure 3C shows UV-Vis spectra of catechin and poly(catechin) powders. The catechin spectrum had two characteristic peaks with maxima at 250 nm and 460 nm. Poly(catechin) powders were characterized by two peaks - with a maximum at 250 nm and a broad peak between 300 nm and 750 nm. Powders obtained during polymerization in pH 7 and pH 8 buffer, both in the dark and during exposure to blue light, showed identical UV-Vis spectra. This indicated the identical structure of all the compounds obtained.

The next step of the research was determining the thermal stability of catechin and polymerization products. Figure 4A–C show the results of thermogravimetry of (+)-catechin and poly(catechin), determined in an inert gas atmosphere—argon. Figure 4D shows the values of T5, T10, and T20 for the substances analyzed, where T5, T10, and T20 refer to the loss of 5%, 10%, and 20%, respectively, of the initial mass of the sample as a function of temperature. The T5, T10 and T20 were determined because further weight loss of the samples was not visible at the measurement conditions (25–800 °C). The (+)-catechin decomposition was two-step, while the polymerized compounds were decomposed in one step. Poly(catechin) obtained during polymerization in buffers at pH 7 and 8, both in the dark and irradiated with blue light, was characterized by a significantly higher thermal resistance than (+)-catechin. The T5 values of poly(catechin) obtained in buffer at pH 7 were about 90 °C higher than T5 of (+)-catechin, and T5 temperatures of poly(catechin) polymerized in buffer at pH 8 were higher by as much as about 160 °C. Moreover, the compounds of polymerized catechin were characterized by low weight loss at 800 °C—the weight loss of poly(catechin) obtained in buffer pH 7 was about 23%, and in buffer pH 8 about 10%. Meanwhile, the reference catechin mass loss at 800 °C was 57%. This meant that the weight loss of poly(catechin) obtained in buffer at pH 7 was about 2.5 times smaller than the standard catechin, and the poly(catechin) polymerized in buffer pH 8 up to about 5.5 times smaller. These results undoubtedly testify to the higher thermal stability of the polymerized compounds. The poly(catechin) obtained in buffer at pH 8 had higher thermal stability than the compounds obtained at pH 7. The phosphate buffer at pH 8 was the environment in which poly(catechin) was obtained with the highest yield, which means that the polymerization product was the most condensed and crosslinked. Condensed catechins—tannins—have high thermal resistance by nature.

According to the literature [40], the complex condensed aromatic structure of natural tannins leads to high thermal resistance. It has been shown that the degradation of Acacia dealbata tannin was almost complete at a temperature of 600 °C, the remaining weight of tannin was approximately 44%, so the weight loss of the sample was 56%. The weight loss of poly (catechin) at 600 °C obtained in buffer at pH 7 (darkness and blue light) was about 22%, the poly (catechin) produced in buffer solution at pH 8 (also darkness and blue light) was about 10%, and of the reference catechin was about 49%. Comparing the results obtained with the literature data on natural condensed tannin, it should be stated that the reference catechin had weaker thermal stability than tannin (which seems obvious), whereas polymeric structures showed better thermal stability than natural tannin (poly (catechin) pH 7 around 2.5 times; poly (catechin) pH 8 about 5.5 times). Besides the condensed structure of tannins, which by nature ensure good thermal resistance, high thermal stability of polymeric complex catechin compounds may also be the result of the presence of phosphorus groups in the powders produced. Phosphorus compounds are recognized as one of the most effective flame retardants used in polymer materials, construction materials, adhesives, etc.

DSC analysis of catechin and poly (catechin) samples was also performed (Figure 5). The (+)-catechin reference sample showed two melting-related endothermic peaks (at 134 °C and 171 °C) and an exothermic oxidation peak at 261 °C. Glass transition temperatures were observed on the curves of polymeric forms of catechin (pH 7 Dark 45 °C, Blue 38 °C; pH 8 Dark 44 °C, Blue 40 °C). The presence of the glass transition on the DSC curve is characteristic of polymeric materials. The poly(catechins) obtained in buffer at pH 7 and 8 had one endothermic peak around 200 °C corresponding to melting. Oxidation temperatures of poly(catechins) samples obtained in phosphate buffer pH 7 were close to the oxidation temperature of (+)-catechin, while oxidation temperatures of polymeric catechins produced in buffer at pH 8 were about 40 °C higher. Poly(catechins) obtained in buffer at pH 8 were characterized by higher resistance to oxidation. Unlike the reference catechin, the oxidation of all poly(catechin) samples was associated with degradation, and the transformations were accompanied by greater enthalpy of oxidation and degradation (ΔH_o_=136–509 J/g) than in the case of catechin oxidation alone (ΔH_o_=13 J/g). Such behavior of the polymeric catechin compound may be due to its higher thermal stability and the shift of the final oxidation temperature within the range of compound decomposition temperatures.

## 4. Conclusions

The first step of polymerization in phosphate buffer at pH 6, 7 and 8 allowed proanthocyanidine to be obtained. The best polymerization effects were achieved in pH 7 and 8 buffer. With regard to literature data [31], it was surprising to obtain proanthocyanidine in the dark, not only under the influence of blue light irradiation. Irradiation of solutions with blue light further accelerated polymerization. Buffer solutions containing poly(catechin) had higher antioxidant activity (marked as hydrogen peroxide scavenging capacity) than the reference (+)-catechin. As a result of the second polymerization step, polymeric complex catechin compound was obtained. The insolubility of powders significantly hindered the determination of the compound structure. However, FTIR analysis indicated the presence of polymeric bonds characteristic of poly(flavonoids) and P-O-C bonds. Thermogravimetric analysis showed high thermal stability of poly(catechin) powders. The compounds of polymerized catechin were characterized by low weight loss at 800 °C—the weight loss of poly(catechin) obtained in buffer pH 7 was about 23%, and in buffer pH 8 about 10%. The polymeric structures of catechin obtained showed better thermal stability than natural tannin—poly(catechin) pH 7 around 2.5 times; poly(catechin) pH 8 about 5.5 times. The natural polymeric compound based on catechin can potentially be used as a pro-ecological stabilizer, e.g., for environmentally friendly materials with wide application.

## Figures and Tables

**Figure 1 biomolecules-10-01191-f001:**
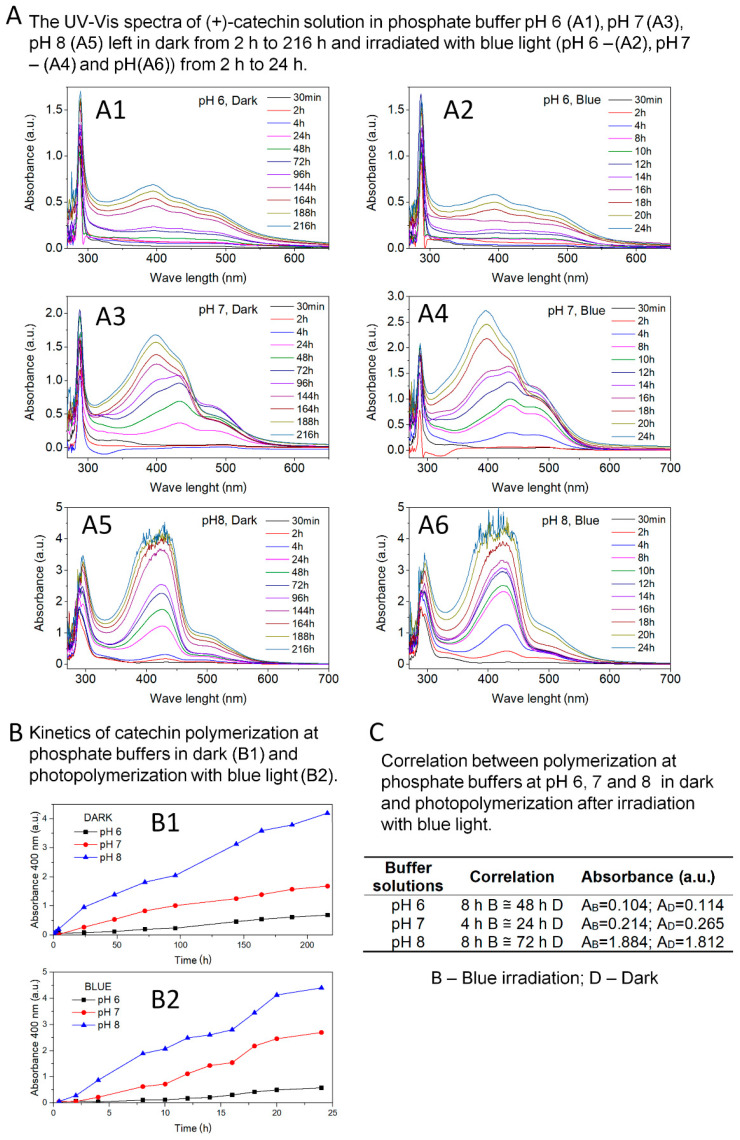
(**A**) The UV-Vis (Ultraviolet–visible) spectra of (+)-catechin solution in phosphate buffer pH 6 (**A1**), pH 7 (**A3**), pH 8 (**A5**) left in the dark from 2 h to 216 h and irradiated with blue light (pH 6—**A2**, pH 7—**A4**, and pH 8—**A6**) from 2 h to 24 h. (**B**) Kinetics of catechin polymerization in phosphate buffers in the dark (**B1**) and photopolymerization after irradiation with blue light (**B2**). (**C**) Correlation between polymerization in phosphate buffers in the dark and photopolymerization after irradiation with blue light.

**Figure 2 biomolecules-10-01191-f002:**
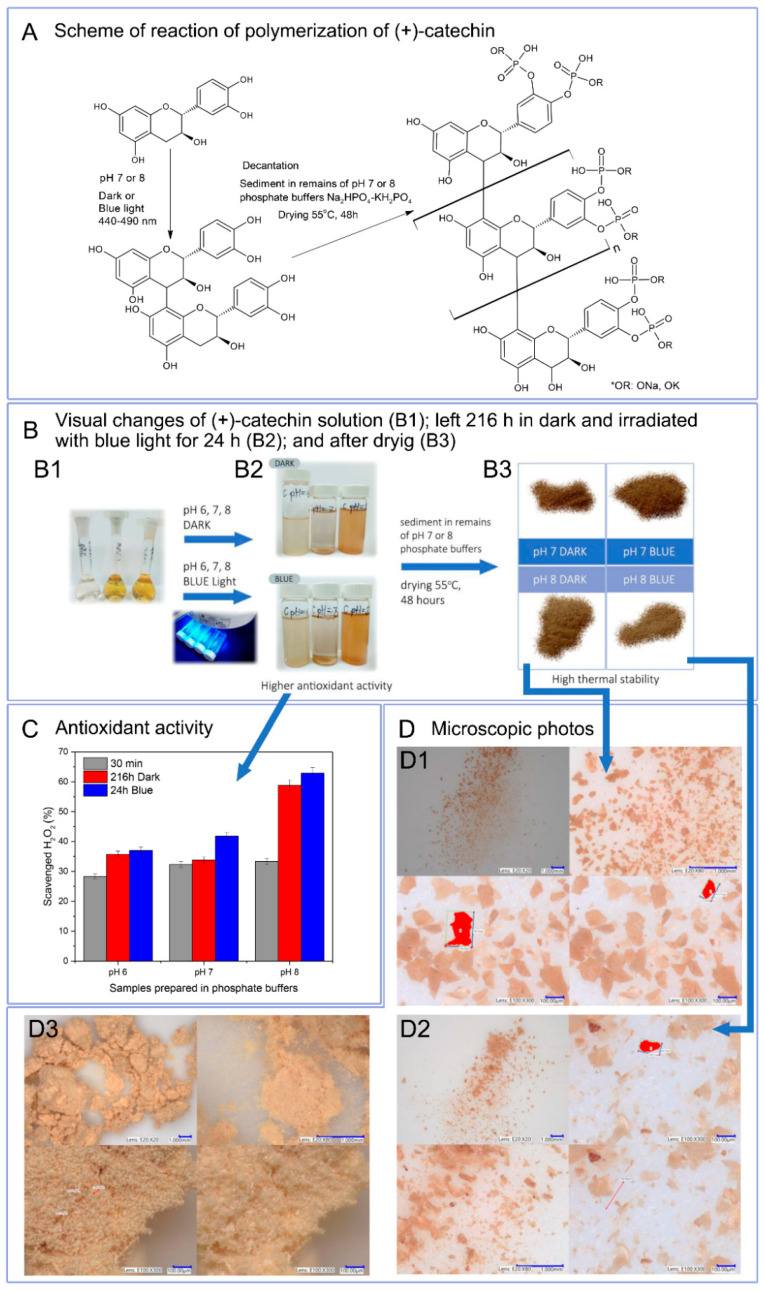
(**A**) Scheme of reaction of photopolymerization of (+)-catechin. (**B**) Visual changes of (+)-catechin solution after preparation (**B1**); left 216 h in the dark and irradiated with blue light for 24 h (**B2**); and after decantation of sediment and drying (**B3**). (**C**) Antioxidant activity of catechin polymerization in phosphate buffers in the dark for 216h and photopolymerization after irradiation with blue light for 24h in phosphate buffer at pH 6, 7, and 8. (**D**) Digital microscopic photos of powder of polymeric complex catechin compound obtained in pH 8 in the dark (**D1**), compound obtained after blue irradiation (**D2**), and reference (+)-catechin (**D3**).

**Figure 3 biomolecules-10-01191-f003:**
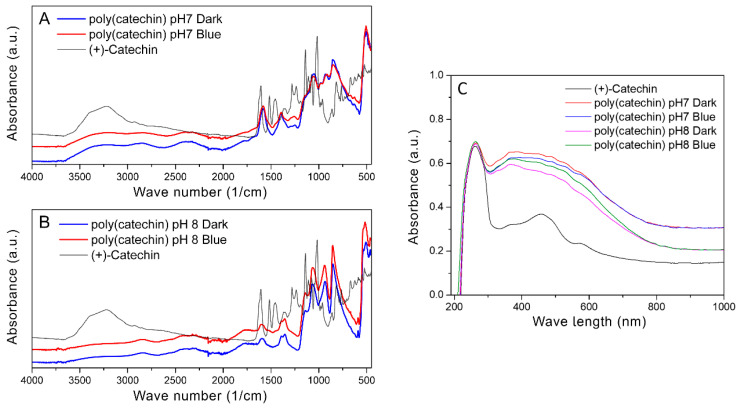
FTIR (Fourier-transform infrared spectroscopy) spectra of poly(catechin) powders obtained as a result of polymerization in the dark in phosphate buffer, and photopolymerization with blue light in solutions of pH 7 (**A**) and pH 8 (**B**) UV-Vis spectra of (+)-catechin and poly(catechin) powders (**C**).

**Figure 4 biomolecules-10-01191-f004:**
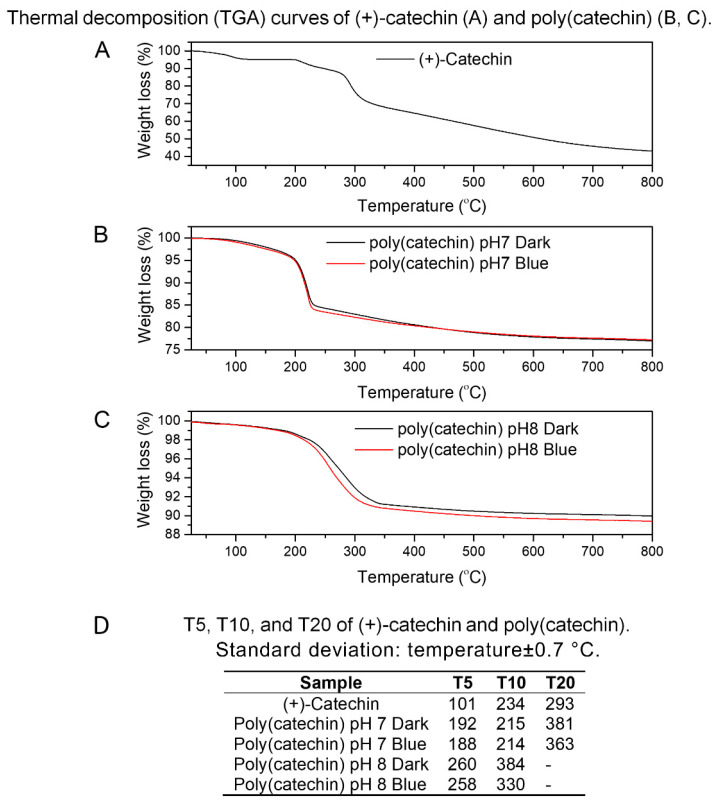
Thermal decomposition (TGA) curves of (+)-catechin (**A**) and poly(catechin) (**B**,**C**). T5, T10, and T20 of (+)-catechin and poly(catechin) (**D**).

**Figure 5 biomolecules-10-01191-f005:**
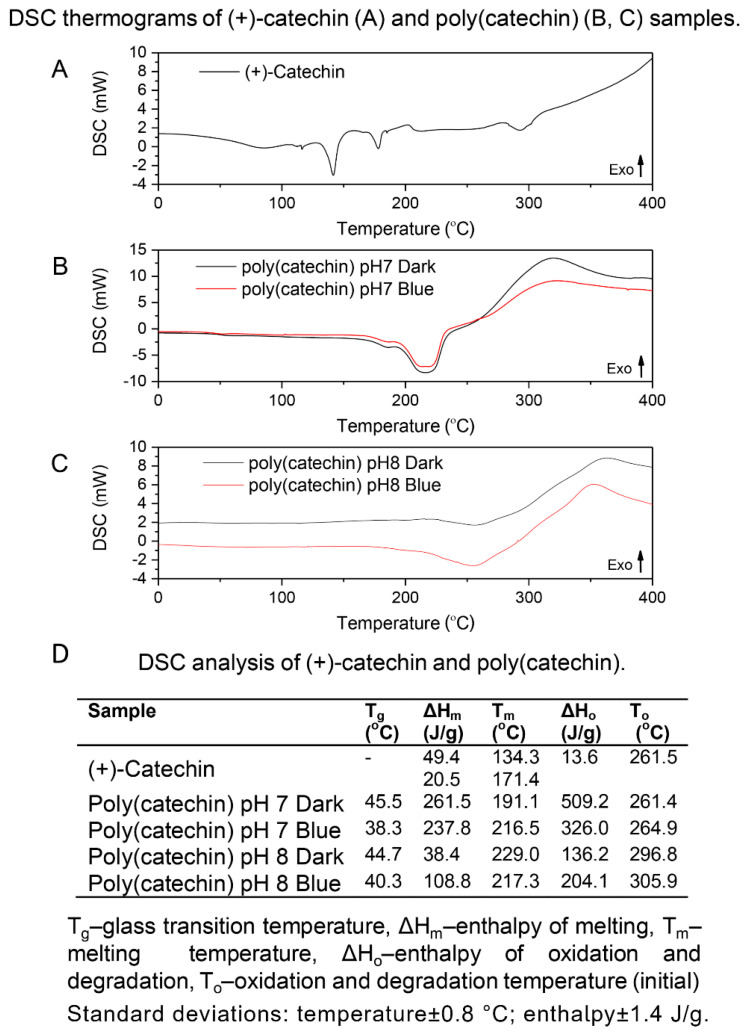
DSC (Differential scanning calorimetry) thermograms of (+)-catechin (**A**) and poly (catechin) (**B**,**C**) samples. DSC analysis of (+)-catechin and poly (catechin) (**D**).

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
