# Peer review of "Natural Polymeric Compound Based on High Thermal Stability Catechin from Green Tea"

_biomolecules, 2020, doi:10.3390/biom10081191_

Round 1

Reviewer 1 Report

In this manuscript, the creation of polymeric catechin that exhibits high thermal stability was presented, an extension of the findings achieved by Liang et al. (2016) with more characterizations on the polymer.  Basically, you guys showed that proanthocyanidins was obtained during the first step, while in the second step, the synthesis of a polymeric complex catechin was achieved.  Both compounds exhibit specific characteristics, i.e., antioxidant activity, and thermal stability, respectively.  The results are sound and well presented.  I personally do not have many objections concerning the publication of the manuscript.  However, there are several points that need to be addressed prior to the acceptance of the manuscript by Biomolecules and are listed as follows.

In Fig. 2A, regarding the dimerization of catechin via the loss of two hydrogen atoms, i.e., an oxidation process, how did the reaction happen?  What is the oxidizing agent for the dimerization that occurs in the dark, if any?  You guys may either propose a reaction mechanism or cite references that may support your findings.

The chemical structure of the polymeric catechin of Fig. 2A needs to be redrawn by a software, e.g., ChemDraw, as the quality of the polymeric catechin with two distorted benzene rings is not suitable for publication.

What are the statistical analyses of all the data in the results section?  For example, in Fig.1B, B1 and B2 can be correlated as their values are very close.  However, how was the numerical value of each data point obtained?  In other words, was each data point an average of three measurements?  If not, the correlation between 4hB and 24hD (at pH7, for example) cannot be expressed as 4hB = 24hD.  Instead, it should be 4hB ≅ 24hD because 4hB ≠ 24hD.  In your future research, you guys are better off expressing your data as mean ± standard deviation (SD) via at least three measurements in order to draw a conclusion that is statistically meaningful.  Additionally, for the thermodynamic data (Tg, ΔHm, Tm,…) in Fig.5, it would be better if they can be expressed as mean ± SD of at least three measurements.

Fig.3C, it should be "length", NOT lenght.

Line 118, as follows, NOT as follow.

For the FTIR analysis, the literature showing the absorption of the functional group, P-O-C, in the range of 1,090-1,000 cm-1 needs to be cited.

Author Response

Institute of Polymer and Dye Technology

Technical University of Lodz

90-924 Lodz, ul Stefanowskiego 12/16, Poland

Tel.: +48 42 631 32 23, Fax: +48 42 636 25 43

August  10, 2020

Biomolecules — Open Access Journal

Dear Professor,

We are resubmitting our revised paper entitled “Natural polymeric compound based on high thermal stability catechin from green tea” by Malgorzata Latos-Brozio and Anna Masek with a request to reconsider it for publication in " Biomolecules”.

We have carefully considered the Editor and Reviewers'  comments. The manuscript was revised exactly according to these comments. The list of responses to the reviewer’s comments and corrections made in the manuscript is attached.

In the manuscript, the changes made based on the Editor's comment are marked in green. Changes relating to Reviewers' comments are marked in red.

The manuscript has not been previously published, is not currently submitted for review to any other journal, and will not be submitted elsewhere before a decision is made by this journal.

For correspondence please use the following information:

corresponding author: Anna Masek

Institute of Polymer and Dye Technology

Technical University of Lodz

90-924 Lodz, ul Stefanowskiego 12/16, Poland

Tel.: +48 42 631 32 93

Fax: +48 42 636 25 43

e-mail: anna.masek@p.lodz.pl

Yours sincerely,

PhD, Dsc Anna Masek

Answers to Reviewer #1 comments

Reviewer #1: In this manuscript, the creation of polymeric catechin that exhibits high thermal stability was presented, an extension of the findings achieved by Liang et al. (2016) with more characterizations on the polymer.  Basically, you guys showed that proanthocyanidins was obtained during the first step, while in the second step, the synthesis of a polymeric complex catechin was achieved.  Both compounds exhibit specific characteristics, i.e., antioxidant activity, and thermal stability, respectively.  The results are sound and well presented.  I personally do not have many objections concerning the publication of the manuscript.  However, there are several points that need to be addressed prior to the acceptance of the manuscript by Biomolecules and are listed as follows.

Reviewer #1: In Fig. 2A, regarding the dimerization of catechin via the loss of two hydrogen atoms, i.e., an oxidation process, how did the reaction happen?  What is the oxidizing agent for the dimerization that occurs in the dark, if any?  You guys may either propose a reaction mechanism or cite references that may support your findings.

Answer: Thank you for your valuable comment.

The formation of catechin dimeric forms was observed both during the irradiation of the solutions with blue light and under the reaction conditions in the dark. Catechin dimerization occurred via the loss of two hydrogen atoms, i.e., an oxidation process. The alkaline reaction environment may be the factor causing the oxidation of catechin in both reactions. Li et al. [1] showed that the stability and degradation of catechins from green tea concentrates was dependent, among others, from the pH of the environment. Green tea catechins are stable under acidic pH conditions. This behavior of catechins can be partly explained by the direct increase in the rate of oxidation with increasing pH. According to the literature data, pKa1 epigallocatechin gallate (EGCG) is 7.55 ± 0.03. Therefore, the ionization state of EGCG, which indicates the ability of the catechin to donate protons, may be a factor in causing catechin ring opening (leading to compound degradation). In a neutral and basic environment, epigallocatechin gallate (EGCG) showed an increased ability to oxidation and the formation of semiquinone free radical of EGCG [1].

Thus, in the reactions described in this manuscript, the oxidizing agent of (+)-catechin during the reaction in the dark as well as during photopolymerization may therefore be an alkaline pH which results in an increased capacity of the catechin to oxidize and form free radicals. These factors contribute to the formation of the dimeric forms of catechin. During polymerization with blue light, the energy of light is additionally a factor accelerating and intensifying the process.

[1] Li, N.; Taylor, L.S.; Ferruzzi, M.G.; Mauer, L.J. Kinetic Study of Catechin Stability: Effects of pH, Concentration, and Temperature. J. Agric. Food Chem. 2012, 60, 12531−12539.

Reviewer #1: The chemical structure of the polymeric catechin of Fig. 2A needs to be redrawn by a software, e.g., ChemDraw, as the quality of the polymeric catechin with two distorted benzene rings is not suitable for publication.

Answer: The chemical structure of the polymeric catechin of Fig. 2A has been corrected.

Reviewer #1: What are the statistical analyses of all the data in the results section?  For example, in Fig.1B, B1 and B2 can be correlated as their values are very close.  However, how was the numerical value of each data point obtained?  In other words, was each data point an average of three measurements?  If not, the correlation between 4hB and 24hD (at pH7, for example) cannot be expressed as 4hB = 24hD.  Instead, it should be 4hB 24hD because 4hB ≠ 24hD.  In your future research, you guys are better off expressing your data as mean ± standard deviation (SD) via at least three measurements in order to draw a conclusion that is statistically meaningful.  Additionally, for the thermodynamic data (Tg, ΔHm, Tm,…) in Fig.5, it would be better if they can be expressed as mean ± SD of at least three measurements.

Answer: We thank the Reviewer for the important comment. We agree with the comment. The correlations in Figure 1 have been corrected (correlations were presented as 4hB ≅ 24hD).

Unfortunately, we did not make three measurements for each sample in the thermodynamic analysis. The reason for this was too small amount of the resulting polymers, which we wanted to test with other methods, apart from DSC and TG. Therefore, in the figures relating to thermal analysis (Fig. 4 - TG and Fig. 5. - DSC) measurement errors of apparatus have been given (Fig 4. Standard deviation: temperature±0.7°C.; Fig 5. Standard deviations: temperature±0.8°C; enthalpy±1.4 J/g.).

In the ‘Hydrogen peroxide scavenging capacity’ assay (Fig. 2), three measurements were made for each sample. The results are shown as mean ± standard deviation (SD).

Reviewer #1: Fig.3C, it should be "length", NOT lenght.

Answer: Fig.3C has been corrected in manuscript.

Reviewer #1: Line 118, as follows, NOT as follow.

Answer: Line 118 has been improved.

Reviewer #1: For the FTIR analysis, the literature showing the absorption of the functional group, P-O-C, in the range of 1,090-1,000 cm-1 needs to be cited.

Answer: We agree with the Reviewer's comment. Literature reference has been added.

Reviewer 2 Report

The authors in the paper tried to improve the thermal stability and antioxidant properties of catechin by polymerization methods under different conditions (neutral, acidic, and alkaline). Catechin can be found widely as cacao and tea constituents, and the research from this paper can potentially provide some useful information on environmentally friendly materials to the researchers in this field. The paper was well organized and written and the scientific results were relatively well proved. I indeed recommend the paper can be accepted after some minor revision.

The questions and comments are as follow:

1. In the abstract, the authors mentioned “This catechin-based compound has not previously been described” which is not accurate since I can easily search for some related literature on this topic such as Pharmaceutical biology 2004, 1, 42, 84, Biomacromolecules 2009, 10(7), 1923, and Macromolecular Bioscience 2003, 3(12), 758. The authors may want to revise this to soften the statement.

2. It seems that the more alkaline or more basic condition is the higher thermal stability of the compound will be, I wonder if the authors tried higher pH than 8.

3. The catechin traces in Figure 3A and 3B are not very clear, the authors may need to change the color in order to visualize the traces.

Author Response

Institute of Polymer and Dye Technology

Technical University of Lodz

90-924 Lodz, ul Stefanowskiego 12/16, Poland

Tel.: +48 42 631 32 23, Fax: +48 42 636 25 43

August  10, 2020

Biomolecules — Open Access Journal

Dear Professor,

We are resubmitting our revised paper entitled “Natural polymeric compound based on high thermal stability catechin from green tea” by Malgorzata Latos-Brozio and Anna Masek with a request to reconsider it for publication in " Biomolecules”.

We have carefully considered the Editor and Reviewers'  comments. The manuscript was revised exactly according to these comments. The list of responses to the reviewer’s comments and corrections made in the manuscript is attached.

In the manuscript, the changes made based on the Editor's comment are marked in green. Changes relating to Reviewers' comments are marked in red.

The manuscript has not been previously published, is not currently submitted for review to any other journal, and will not be submitted elsewhere before a decision is made by this journal.

For correspondence please use the following information:

corresponding author: Anna Masek

Institute of Polymer and Dye Technology

Technical University of Lodz

90-924 Lodz, ul Stefanowskiego 12/16, Poland

Tel.: +48 42 631 32 93

Fax: +48 42 636 25 43

e-mail: anna.masek@p.lodz.pl

Yours sincerely,

PhD, Dsc Anna Masek

Answers to Reviewer #2 comments

Reviewer #2: The authors in the paper tried to improve the thermal stability and antioxidant properties of catechin by polymerization methods under different conditions (neutral, acidic, and alkaline). Catechin can be found widely as cacao and tea constituents, and the research from this paper can potentially provide some useful information on environmentally friendly materials to the researchers in this field. The paper was well organized and written and the scientific results were relatively well proved. I indeed recommend the paper can be accepted after some minor revision. The questions and comments are as follow:

Reviewer #2: 1. In the abstract, the authors mentioned “This catechin-based compound has not previously been described” which is not accurate since I can easily search for some related literature on this topic such as Pharmaceutical biology 2004, 1, 42, 84, Biomacromolecules 2009, 10(7), 1923, and Macromolecular Bioscience 2003, 3(12), 758. The authors may want to revise this to soften the statement.

Answer: We agree with the Reviewer's comment. Sentence “This catechin-based compound has not previously been described” has been deleted.

Reviewer #2: 2. It seems that the more alkaline or more basic condition is the higher thermal stability of the compound will be, I wonder if the authors tried higher pH than 8.

Answer: We have not tested photopolymerization at pH higher than 8. We thank the Reviewer for the important suggestion. We believe that it is worth extending our research in the future and perform catechin photopolymerization in alkaline buffers with a pH higher than 8.

Reviewer #2: 3. The catechin traces in Figure 3A and 3B are not very clear, the authors may need to change the color in order to visualize the traces.

Answer: We agree with the comment. Figure 3A and 3B have been improved.